# Dissection of the interaction between the intrinsically disordered YAP protein and the transcription factor TEAD

Yannick Mesrouze, Fedir Bokhovchuk, Marco Meyerhofer, Patrizia Fontana, Catherine Zimmermann, Typhaine Martin, Clara Delaunay, Dirk Erdmann, Tobias Schmelzle, Patrick Chène*

Disease Area Oncology, Novartis Institutes for Biomedical Research, Basel, Switzerland

**Abstract** TEAD (*TEA*/ATTS *d*omain) transcription factors are the most distal effectors of the Hippo pathway. YAP (*Yes*-*a*ssociated *p*rotein) is a coactivator protein which, upon binding to TEAD proteins, stimulates their transcriptional activity. Since the Hippo pathway is deregulated in various cancers, designing inhibitors of the YAP:TEAD interaction is an attractive therapeutic strategy for oncology. Understanding the molecular events that take place at the YAP:TEAD interface is therefore important not only to devise drug discovery approaches, but also to gain knowledge on TEAD regulation. In this report, combining single site-directed mutagenesis and double mutant analyses, we conduct a detailed analysis on the role of several residues located at the YAP:TEAD interface. Our results provide quantitative understanding of the interactions taking place at the YAP:TEAD interface and give insights into the formation of the YAP:TEAD complex and more particularly on the interaction between TEAD and the Ω-loop found in YAP.

*For correspondence: patrick_chene@yahoo.com

## Introduction

The Hippo signaling pathway controls organ size and tissue homeostasis in animals (*Zhao et al., 2011*; *Yu et al., 2015*). This pathway is deregulated in various human cancers, and compounds that modulate its activity might have promise as new anticancer agents (*Harvey et al., 2013*; *Gong and Yu, 2015*; *Ye and Eisinger-Mathason, 2016*). The TEAD (*TEA*/ATTS *d*omain, TEAD 1–4) transcription factors are the most downstream effectors of the Hippo pathway, and several coactivator proteins interact with them, regulating their transcriptional activity (*Pobbati and Hong, 2013*). Amongst these coactivators, the YAP (*Yes*-*a*ssociated *p*rotein) protein is often overexpressed in cancers, and targeting the YAP:TEAD interaction is emerging as a new therapeutic strategy in oncology (*Gong and Yu, 2015*; *Felley-Bosco and Stahel, 2014*; *Santucci et al., 2015*; *Zhang et al., 2015*). As a consequence, the study of the YAP:TEAD interaction is important not only to support drug discovery activities, but also to understand the regulation of TEAD transcription factors.

The TEAD binding site of mouse YAP (mYAP) was initially mapped to the N-terminus of this protein (mYAP$^{32\text{-}139}$) (*Vassilev et al., 2001*) and later on more precise knowledge was gained when the structure of the YAP:TEAD complex was published (*Chen et al., 2010a*; *Li et al., 2010*). In these studies, large protein fragments, hYAP$^{50\text{-}171}$ (human YAP) or mYAP$^{35\text{-}92}$, were used for crystallization, but electron density was only observed for the hYAP$^{50\text{-}100}$ and mYAP$^{47\text{-}85}$ regions. Subsequent binding studies showed that hYAP$^{50\text{-}171}$ and hYAP$^{50\text{-}99}$ have a similar affinity for TEAD (*Hau et al., 2013*) suggesting that the regions outside the fragment YAP$^{50\text{-}99}$ have little importance for the interaction. Bound hYAP$^{50\text{-}100}$ contains from the N- to C-terminus: a β-strand, an α-helix, a long loop and an Ω-loop. The study of hYAP$^{2\text{-}268}$ by nuclear magnetic resonance reveals that it is natively unfolded in

solution (*Tian et al., 2010*) and the online ANCHOR software (*Dosztányi et al., 2009*) predicts hYAP[50-100] to be disordered (data not shown). This together with the structural data available on the bound form of YAP indicates that hYAP[50-100] becomes structured upon binding to TEAD. In contrast, the comparison of the free and bound forms of TEAD shows that it is permanently folded and does not undergo large conformational changes upon binding to YAP (the root mean square deviation of atomic positions between TEAD in the YAP:TEAD1 complex (pdb 3KYS) and apo TEAD3 (pdb 5EMW) is 0.443 Å).

The contribution to binding of the β-strand from the TEAD-binding site of YAP is modest (*Hau et al., 2013*) and mainly due to the formation of hydrogen bonds between the main chain atoms of YAP and TEAD (*Li et al., 2010*). Studies with synthetic peptides mimicking the α-helix and the Ω-loop regions of YAP show that only Ω-loop mimetics have a weak (around 70 μM) but measurable affinity for TEAD (*Hau et al., 2013*; *Pobbati et al., 2015*). Data obtained from cellular assays reveal that mutations in the Ω-loop have a greater impact on the formation of the YAP:TEAD complex than mutations of α-helix residues (*Chen et al., 2010a*; *Li et al., 2010*). Similarly, mutations of TEAD in its Ω-loop binding pocket affect the interaction with YAP to a greater extent than mutations in its α-helix binding pocket (*Chen et al., 2010a*; *Li et al., 2010*; *Tian et al., 2010*).

Overall this indicates that YAP interacts with TEAD mainly via two distinct secondary structure elements, an α-helix and an Ω-loop, and that the latter contributes most to the formation of the YAP:TEAD complex. However, the majority of the data available today provide only semi-quantitative information on the contribution to binding of the residues located at the YAP:TEAD interface. To better understand how these two proteins interact, a more quantitative knowledge needs to be developed. In this report, we conduct a detailed analysis of the YAP:TEAD interface and examine the contribution of several key residues located on both proteins to the formation of the YAP:TEAD complex. This work, which is based on an analysis of the effect of single mutations and on the coupling energies measured from double mutant cycles, gives a new insight into the energetics of the events taking place during the association between YAP and TEAD.

## Results and dicussion

### Mutations in the α-helix and Ω-loop regions of hYAP[50-171]

Analysis of the structure of different YAP:TEAD complexes (pdb 3KYS and 3JUA) (*Chen et al., 2010a*; *Li et al., 2010*), the data from cellular assays (*Chen et al., 2010a*; *Li et al., 2010*; *Tian et al., 2010*) and the results obtained from structure function studies carried out with synthetic peptides (*Hau et al., 2013*; *Zhang et al., 2014*; *Mesrouze et al., 2016*) allowed us to select several YAP residues that should contribute significantly to the interaction with TEAD. In the α-helix (hYAP[61-73]) region, hYAP Leu65, Leu68, and Phe69 belong to the LxxLF motif, which is known to bind to hydrophobic grooves (*Santucci et al., 2015*; *Li et al., 2010*). In the YAP:TEAD complex, these three residues are located on the same side of the α-helix making hydrophobic interactions with TEAD (*Figure 1A*). In the Ω-loop region (hYAP[85-99]), hYAP Met86, Leu91, and Phe95 are involved in hydrophobic interactions with TEAD, and they interact with each other to form a hydrophobic core within the bound Ω-loop (*Figure 1B*). hYAP Arg89 interacts with hTEAD4 Asp272 and Gln269 (*Figure 1C*) while hYAP Ser94 is within hydrogen bond distance from hTEAD4 Glu263 and Tyr429 (*Figure 1D*). The amino acid hYAP Phe96 does not provide any direct interaction with TEAD. It is located above the hydrophobic core formed by hYAP Met86, Leu91 and Phe95 (*Figure 1B*) and it interacts also with the guanidinium group of hYAP Arg87 (not shown, but see *Zhang et al. [2014]*). This residue was included in this study because it may contribute to the stabilization of the bound Ω-loop.

These 9 YAP residues were mutated to alanine. The mutations were carried out in hYAP[50-171], which contains the TEAD-binding domain and shows a good affinity for TEAD (*Li et al., 2010*; *Hau et al., 2013*). The affinity (dissociation constant, $K_d$) of these proteins was measured by Surface Plasmon Resonance (SPR) in experiments where N-Avitagged hTEAD4[217-434] was immobilized on sensor chips. The $K_d$ of wt hYAP[50-171] (18 nM, *Table 1*) is similar to the affinity of the longer hYAP[2-268] ($K_d$ = 33 nM, [*Tian et al., 2010*]) showing that all the residues needed for the interaction with TEAD are probably present in hYAP[50-171].

Each of the nine single alanine mutations of hYAP[50-171] significantly destabilizes the YAP:TEAD complex (*Table 1*) confirming the role of the studied residues in the interaction. The hYAP Leu68Ala

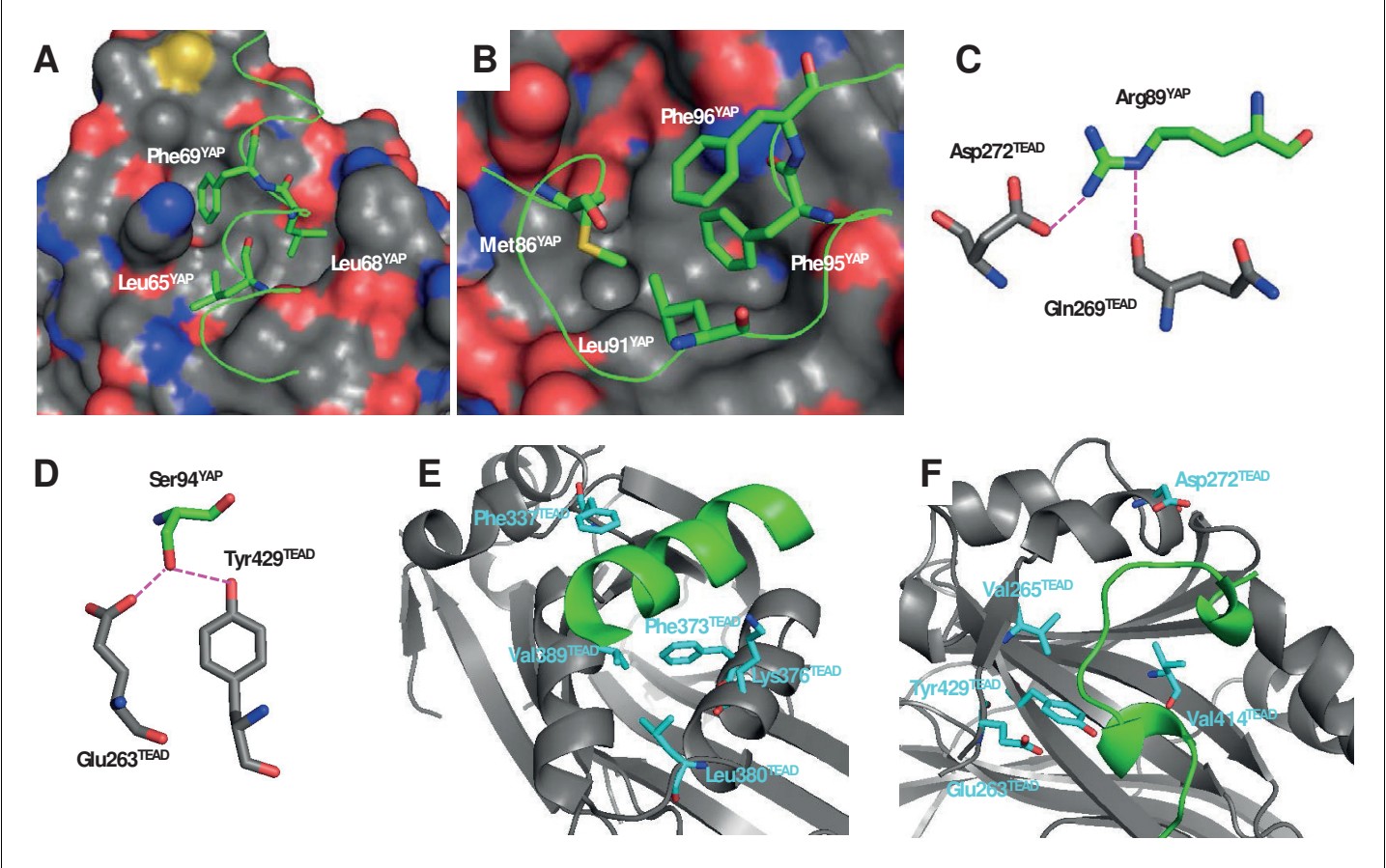

**Figure 1.** Structure of the YAP:TEAD complex. The α-helix (A) and Ω-loop (B) binding interfaces. The surface of TEAD is represented and green ribbons indicate the α-helix (A, region 61–73) or the Ω-loop (B, region 85–99) of YAP. The different YAP residues that have been mutated are indicated. Interactions between hYAP Arg89 and TEAD (C) and between hYAP Ser94 and TEAD (D). The hydrogen bonds are represented by dotted purple lines. The TEAD α-helix (E) and Ω-loop (F) binding pockets. TEAD and YAP are represented by gray and green ribbons, respectively. The mutated TEAD residues for which a $K_d^{eq}$ has been measured are represented in cyan. These figures are drawn from the pdb structure 3KYS (*Li et al., 2010*). TEAD residues are labeled according to hTEAD4 primary sequence.

mutation is the least destabilizing ($\Delta\Delta G = \Delta G_{mutant} - \Delta G_{wt} = 1.92$ kcal/mol) and the Leu91Ala mutation is the most destabilizing ($\Delta\Delta G = 4.4$ kcal/mol). In the α-helix region, the hYAP Phe69Ala mutation has the greatest effect on binding ($\Delta\Delta G = 3.48$ kcal/mol). This residue, which is strictly conserved both in the YAP proteins and in the other TEAD coactivators TAZ and VGLL, is therefore key for the interaction between the α-helix region of YAP and TEAD. All the mutations in the Ω-loop region have a major effect on the YAP:TEAD interaction, reducing the binding energy by more than 2.5 kcal/mol. However, great caution is warranted in interpreting these data. In contrast to hYAP Leu65, Leu68 and Phe69, which show few (if any) intramolecular interactions with each other in the bound α-helix (*Figure 1A*), hYAP Met86, Leu91, Phe95, and Phe96 show intramolecular hydrophobic interactions that may help in stabilizing the bound Ω-loop. Alanine mutations of these residues may disturb these stabilizing interactions in addition to disrupting contacts with TEAD. The results obtained with hYAP Phe96 are a good example of the impact that intramolecular interactions have on the binding of the Ω-loop. This residue is not in direct contact with TEAD, and it is above the hydrophobic core of the bound Ω-loop formed by hYAP Met86, Leu91, and Phe95 (*Figure 1B*). The large decrease in binding ($\Delta\Delta G > 3$ kcal/mol, *Table 1*) measured for hYAP Phe96Ala is therefore due not to the loss of interactions with TEAD, but most probably to a destabilization of the bound Ω-loop. Mutations of residues located in the hydrophobic core of the Ω-loop may therefore affect both intramolecular and intermolecular interactions, explaining why they are often more destabilizing than

**Table 1.** Binding affinities of the different YAP$^{50-151}$ proteins for wt hTEAD4$^{217-434}$. The affinities were measured at 298°K by Surface Plasmon Resonance in n ≥ 3 independent experiments. N-Avitagged hTEAD4$^{217-434}$ was immobilized on sensor chips. The $K_d$ values were obtained from equilibrium data ($K_d^{eq}$). Averages and standard errors (SE) are given. $\Delta\Delta G = \Delta G_{mutant} - \Delta G_{wt}$ and $SE_{\Delta\Delta G} = (SE_{\Delta Gmutant}^2 + SE_{\Delta Gwt}^2)^{1/2}$.

| | YAP$^{50-171}$ | $K_d^{eq}$ (nM) | $\Delta G$ (kcal/mol) | $\Delta\Delta G$ (kcal/mol) |
|---|---|---|---|---|
| **Mutation site** | **Wild type** | **18 ± 0** | **−10.56 ± 0.01** | |
| **α-helix** | Leu65Ala | 794 ± 4 | −8.31 ± 0.01 | 2.24 ± 0.01 |
| | Leu68Ala | 464 ± 4 | −8.63 ± 0.01 | 1.92 ± 0.01 |
| | Phe69Ala | 6447 ± 400 | −7.08 ± 0.04 | 3.48 ± 0.04 |
| **Ω-loop** | Met86Ala | 2080 ± 69 | −7.74 ± 0.02 | 2.81 ± 0.02 |
| | Arg89Ala | 27423 ± 381 | −6.22 ± 0.01 | 4.34 ± 0.01 |
| | Leu91Ala | 30550 ± 2250 | −6.15 ± 0.04 | 4.40 ± 0.05 |
| | Ser94Ala | 5623 ± 341 | −7.16 ± 0.04 | 3.40 ± 0.04 |
| | Phe95Ala | 26045 ± 755 | −6.25 ± 0.02 | 4.31 ± 0.02 |
| | Phe96Ala | 4755 ± 245 | −7.25 ± 0.03 | 3.30 ± 0.03 |

the mutations in the α-helix. The hYAP Arg89Ala and Ser94Ala mutations trigger a large decrease in affinity ($\Delta\Delta G > 3$ kcal/mol). This is in agreement with earlier work showing the importance of the interactions made between these two amino acids and TEAD (*Li et al., 2010*; *Hau et al., 2013*; *Zhang et al., 2014*).

## Mutations in the α-helix and Ω-loop binding pockets of hTEAD4$^{217-434}$

The structure of the YAP:TEAD complex (pdb 3KYS) was analyzed to find the TEAD residues that are located in the surrounding (within 5 Å) and have the potential to interact with the 9 YAP amino acids studied above. This led to the identification of 15 TEAD residues (*Table 2*) that were mutated to determine their role in the formation of the YAP:TEAD complex. In contrast to the TEADbinding domain of YAP, which is natively unfolded in solution, the free form of TEAD is folded. Mutations of TEAD may therefore affect its structure, making an analysis of the role played in the interaction by the mutated residues more challenging. To estimate the impact of the mutations on hTEAD4$^{217-434}$, the mutant proteins were studied in a fluorescence-based thermal shift assay, a technique commonly used to assess the effect of mutations on proteins (see for example, [*Lavinder et al., 2009*; *Bultema et al., 2014*; *Decroos et al., 2015*; *Rumora et al., 2016*]). The melting temperature ($T_m$) of the 15 TEAD mutant proteins was measured, and their difference in thermal stability compared with that of wt hTEAD4$^{217-434}$ was calculated: $\Delta T_m = T_m^{wt} - T_m^{mutant}$ (*Table 2*). The mutations reduced the thermal stability of hTEAD4$^{217-434}$ between 0.1°C and 6.6°C. Using $\Delta T_m > 4$°C as cut-off value, the five mutations hTEAD4 Lys273Ala, Leu295Ala, Lys297Ala, Tyr369Ala, and Leu377Ala were considered too destabilizing, and the corresponding proteins were not further studied. We shall assume in the following that the other mutations did not substantially affect the overall TEAD structure. The affinity of these proteins for wt hYAP$^{50-171}$ was measured by SPR. The localization of these residues at TEAD surface is given on *Figure 1E and F*.

The hTEAD4 Phe373Ala (α-helix pocket) and Val414Ala (Ω-loop pocket) mutations lead to a less than twofold reduction in the affinity of hYAP$^{50-171}$, showing that the contribution of these residues to the interaction is minimal (*Table 2*). The corresponding mutations in hTEAD1 or mTEAD4 also have a modest effect in cell-based assays (*Li et al., 2010*; *Chen et al., 2010b*).

Within the Ω-loop binding pocket, hYAP Ser94 makes a hydrogen bond with hTEAD4 Glu263 and Tyr429 (*Figure 1D*) and the hTEAD4 Glu263Ala and Tyr429Phe mutations destabilize binding with $\Delta\Delta G = 1.48$ and 0.81 kcal/mol, respectively (*Table 2*). This reveals that the loss of the hydrogen bond between hYAP Ser94 and hTEAD4 Glu263 has a greater impact on binding than the loss of the hydrogen bond between hYAP Ser94 and hTEAD4 Tyr429. The mutation corresponding to hTEAD4

**Table 2.** Properties of the different hTEAD4$^{217-434}$ proteins. The melting temperatures ($T_m$) of the proteins were determined in n $\geq$ 3 independent experiments in a fluorescence-based thermal shift assay. Averages and standard errors (SE) are given. $\Delta T_m = T_m^{mutant} - T_m^{wt}$ and $SE_{\Delta Tm} = (SE_{Tmmutant}^2 + SE_{Tmwt}^2)^{1/2}$. For dissociation constant measurements, the different N-Avitagged hTEAD4$^{217-434}$ proteins were immobilized on sensor chips and their affinity for wt hYAP$^{50-171}$ was measured at 298°K by Surface Plasmon Resonance in n $\geq$ 3 independent experiments. $K_d$ values were obtained from equilibrium data ($K_d^{eq}$). Averages and standard errors (SE) are given. $\Delta \Delta G = \Delta G_{mutant} - \Delta G_{wt}$ and $SE_{\Delta \Delta G} = (SE_{\Delta Gmutant} + SE_{\Delta Gwt})^{1/2}$. n. d.: not determined.

| | hTEAD4$^{217-434}$ | $T_m$ (°C) | $\Delta T_m$ (°C) | $K_d^{eq}$ (nM) | $\Delta G$ (kcal/mol) | $\Delta \Delta G$ (kcal/mol) |
|---|---|---|---|---|---|---|
| Mutation site | Wild type | 54.0 ± 0.1 | | 18 ± 0 | −10.56 ± 0.01 | |
| α-helix | Phe337Ala | 53.1 ± 0.1 | −0.9 ± 0.1 | 202 ± 8 | −9.12 ± 0.02 | 1.43 ± 0.02 |
| | Tyr369Ala | 49.9 ± 0.2 | −4.2 ± 0.3 | n. d. | | |
| | Phe373Ala | 53.0 ± 0.3 | −1.0 ± 0.3 | 12 ± 0 | −10.79 ± 0.00 | −0.24 ± 0.01 |
| | Lys376Ala | 53.4 ± 0.1 | −0.7 ± 0.1 | 203 ± 9 | −9.13 ± 0.03 | 1.43 ± 0.03 |
| | Leu377Ala | 49.8 ± 0.1 | −4.3 ± 0.2 | n. d. | | |
| | Leu380Ala | 52.8 ± 0.1 | −1.3 ± 0.2 | 162 ± 7 | −9.26 ± 0.03 | 1.30 ± 0.03 |
| | Val389Ala | 53.0 ± 0.0 | −1.0 ± 0.1 | 304 ± 5 | −8.88 ± 0.01 | 1.67 ± 0.01 |
| Ω-loop | Glu263Ala | 53.0 ± 0.2 | −1.1 ± 0.2 | 220 ± 9 | −9.07 ± 0.02 | 1.48 ± 0.03 |
| | Val265Ala | 52.0 ± 0.0 | −2.0 ± 0.1 | 106 ± 3 | −9.51 ± 0.02 | 1.05 ± 0.02 |
| | Asp272Ala | 54.1 ± 0.1 | 0.1 ± 0.1 | 6995 ± 317 | −7.03 ± 0.03 | 3.53 ± 0.03 |
| | Lys273Ala | 48.1 ± 0.2 | −5.9 ± 0.3 | n. d. | | |
| | Leu295Ala | 47.4 ± 0.1 | −6.6 ± 0.1 | n. d. | | |
| | Lys297Ala | 48.4 ± 0.2 | −5.7 ± 0.3 | n. d. | | |
| | Val414Ala | 50.9 ± 0.1 | −3.2 ± 0.1 | 14 ± 1 | −10.73 ± 0.02 | −0.17 ± 0.02 |
| | Tyr429Phe | 52.3 ± 0.1 | −1.7 ± 0.1 | 71 ± 6 | −9.75 ± 0.05 | 0.81 ± 0.05 |

Glu263Ala in hTEAD1 (Glu240) also has a significant effect on the YAP:TEAD interaction in cell-based assays (*Li et al., 2010*). In cells, the change to alanine/histidine in hTEAD1-2 or mTEAD4 of the tyrosine residue corresponding to hTEAD4 Tyr429 markedly reduces the interaction between YAP and TEAD (*Chen et al., 2010a*; *Li et al., 2010*; *Tian et al., 2010*). These alanine/histidine mutations are therefore more destabilizing than the phenylalanine mutation studied here. The hTEAD1 Tyr421His mutation (equivalent to hTEAD4 Tyr429His) is also detrimental in a more physiologic context, since it is associated with Sveinsson's chorioretinal atrophy (*Fossdal et al., 2004*). Bringing together the published data with our results suggests that the histidine mutation probably has additional effects on the YAP:TEAD interaction beyond just preventing the formation of a hydrogen bond with hYAP Ser94.

The residues hTEAD4 Phe337, Lys376, Leu380, and Val389 are located in the α-helix binding pocket of TEAD, making hydrophobic contacts with bound YAP. hTEAD4 Phe337 is in the vicinity of hYAP Leu68. hTEAD4 Lys376 is close to hYAP Leu65 and Phe69. hTEAD4 Leu380 and Val389 are within van de Waals distance from hYAP Phe69. The mutation of these residues to alanine decreases $\Delta G$ by more than one kcal/mol (*Table 2*), showing that they contribute to the formation of the YAP:TEAD complex. Some of these mutations (Phe337Ala and Leu380Ala) also affect the interaction between YAP and TEAD in cell-based assays when carried out at equivalent positions in mTEAD4 (*Chen et al., 2010a*).

In the Ω-loop binding pocket, the mutation of hTEAD4 Val265, which is in the vicinity of hYAP Leu91, has a significant effect on YAP binding ($\Delta \Delta G \sim$ 1 kcal/mol, *Table 2*). The mutation of the equivalent residue in hTEAD1 has a modest effect in cells on the YAP:TEAD interaction (*Li et al.,*

*2010*). The Asp272Ala mutation has a major impact on the formation of the YAP:TEAD complex with $\Delta\Delta G$ > 3.5 kcal/mol (*Table 2*). Similar results were obtained with different hTEAD4 Asp272Ala protein preparations (data not shown). The mutation does not modify the $T_m$ of hTEAD4$^{217\text{-}434}$ ($\Delta T_m$ = 0.1°C, *Table 2*) and it does not significantly affect hTEAD4 structure since the circular dichroism spectra of hTEAD4 Asp272Ala and wt hTEAD4 are identical (*Figure 2—figure supplement 1*). In the available structures of unbound TEAD (pdb 3L15 [*Tian et al., 2010*], pdb 5EMV and 5EMW [*Noland et al., 2016*], pdb 5HGU [*Chan et al., 2016*]), this residue is exposed to the solvent and does not seem to be involved in well-defined intramolecular interactions (data not shown). In the YAP:TEAD complex, it only interacts via its carboxylic group with the guanidinium moiety of hYAP Arg89 (*Figure 1C*). Since the hYAP Arg89Ala mutation also significantly reduces binding (*Table 1* and [*Hau et al., 2013*]), the interaction between these two residues appears to be key for the formation of the YAP:TEAD complex. The role of hYAP Arg89 in the formation of the YAP:TEAD complex has been demonstrated in cell-based assays (*Li et al., 2010*). However, hTEAD4 Asp272 (or the corresponding residue in the TEAD 1–3) has not been studied in the different structure-function analyses known to us (*Chen et al., 2010a*; *Li et al., 2010*; *Tian et al., 2010*). To determine if this residue is also important for formation of the YAP:TEAD complex in cells in the context of the full length proteins, we used two different cellular assays. The effect of the hTEAD4 Asp272Ala mutation was assessed in co-immunoprecipitation experiments (*Figure 2A*). N-terminally V5-tagged hTEAD4 (wt or Asp272Ala) and wt hYAP were co-transfected into HEK293FT cells. YAP was immunoprecipitated and the TEAD4 protein complexed to it was detected by Western blot. While wt TEAD4 co-immuno-precipitates with YAP, the hTEAD4 Asp272Ala mutant does not to detectable levels. This shows that the hTEAD4 Asp272Ala mutation prevents the formation of the YAP:TEAD complex in cells. To confirm this result in a more functional assay, we used a YAP:TEAD-responsive gene reporter assay

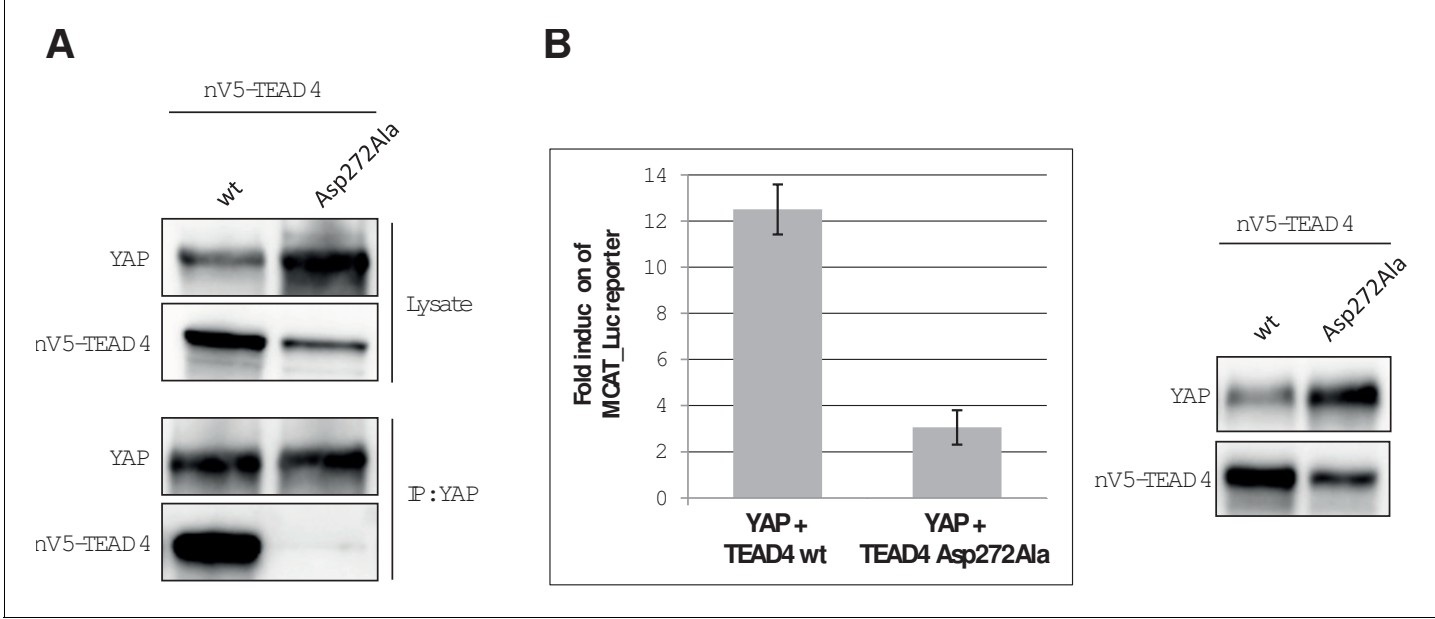

**Figure 2.** Effect of the hTEAD4 Asp272Ala mutation in cells. (**A**) Co-immunoprecipitation: N-terminally V5-tagged hTEAD4 (wild-type (wt) or Asp272Ala mutant) were co-transfected with wt hYAP into HEK293FT cells. YAP was immunoprecipitated, and co-immunoprecipitated nV5-TEAD4 was determined by anti-V5 Western Blot. (**B**) MCAT_Luc reporter assay: N-terminally V5-tagged TEAD4 (wt or Asp272Ala mutant) were co-transfected with wt hYAP into the HEK293::MCAT_Luc reporter model. Resazurin-normalized luciferase activity was measured and is plotted as fold induction over baseline. YAP and nV5-TEAD expression levels were determined in parallel by Western Blot. The expression level of hTEAD4 Asp272Ala mutant versus wt hTEAD4 was quantified by Image J software to be reduced by a factor of approx. 1.6, indicating that the approximate fourfold reduction in MCAT_Luc reporter signal cannot be solely attributed to expression differences, but does truly reflect a reduced activity of the hTEAD4 Asp272Ala mutant.

The following figure supplement is available for figure 2:

**Figure supplement 1.** Analysis of the structure of the unbound hTEAD4 Asp272Ala mutant protein by Circular Dichroism (CD).

where the expression of the luciferase gene is under the control of multiple MCAT sites, which function as TEAD recognition sequences (*Michaloglou et al., 2013*) (*Figure 2B*). The transfection of wt hYAP together with wt TEAD4 in HEK293::MCAT cells stimulates the MCAT-based reporter, leading to marked luciferase activity. Under the same conditions, induction of luciferase activity is significantly reduced when hYAP is co-transfected with hTEAD4 Asp272Ala. This indicates that the mutation compromises the ability of YAP to stimulate TEAD4 transcriptional activity. These cellular data demonstrate that the hTEAD4 Asp272 mutation affects the interaction between the full-length YAP and TEAD4 proteins in a cellular context. Together with the published data on hYAP Arg89 (*Li et al., 2010*) this demonstrates that the hYAP Arg89:hTEAD4 Asp272 interaction is also key for the formation of the YAP:TEAD complex in a cellular environment.

## Double mutant cycle analyses

Single mutations allow the identification of key residues involved in the interaction between two proteins (*DeLano, 2002*; *Kortemme and Baker, 2002*). However, they do not serve to determine whether (or not) two residues act in a cooperative manner in the interaction (*Kortemme et al., 2004*). This information is provided by double mutant cycle analyses (DMCAs) (*Mildvan et al., 1992*; *Horovitz, 1996*). DMCAs, which were initially conducted to study interactions within a protein (*Carter et al., 1984*), have been extended to study protein-protein interactions (for example, [*Schreiber and Fersht, 1995*; *Goldman et al., 1997*; *Bradshaw and Waksman, 1999*; *Kiel et al., 2004*; *Naider et al., 2007*; *Elliot-Smith et al., 2007*; *Gianni et al., 2011*; *Jemth et al., 2014*]). They allow the coupling (or interaction) energy, $\Delta\Delta G_{int}$ (see Materials and methods section for details), between two residues at the interface of a protein complex to be calculated. The two residues are independent, and their effect on binding energy is additive when $\Delta\Delta G_{int} = 0$. In the other cases ($\Delta\Delta G_{int} \neq 0$), they are energetically coupled (they exert an energetic effect on each other), and coupling is favorable when $\Delta\Delta G_{int} < 0$ or unfavorable when $\Delta\Delta G_{int} > 0$.

The YAP:TEAD interface is a particularly attractive model to carry out DMCA since it is formed of two distant elements. DMCA should allow us not only to gain more insight into the interactions taking place within each of these pockets, but also to determine whether residues from the two different pockets are energetically coupled. The YAP and TEAD mutations that destabilize the hYAP$^{50-171}$: hTEAD4$^{217-434}$ interaction by more than 1 kcal/mol were selected for this analysis. The pairwise study of these different YAP and TEAD proteins led to the measurement of 80 different $K_d$ values (*Supplementary file 1*), which enabled us to construct 63 double mutant cycles (*Figure 4—source data 1*).

The $\Delta\Delta G_{int}$ values obtained from this analysis are given in *Table 3*. We shall start by studying pairs of residues that are not located at the same binding interface. $\Delta\Delta G_{int}$ could not be determined for 7 of such pairs (n. m. in *Table 3*) because the $K_d$ values for the mutantYAP:mutant TEAD interaction fall outside the scope of our experiment (>200 µM, *Supplementary file 1*). For the remaining pairs, most of the $\Delta\Delta G_{int}$ values are within experimental error ($\Delta\Delta G_{int} = 0 \pm 0.5$ kcal/mol, see Materials and methods). This means that the residues in these pairs are not energetically coupled and that they act independently of the formation of the YAP:TEAD complex. Earlier work on long-range effects on the additivity of mutations showed that $\Delta\Delta G_{int}$ decreases with the distance between the two mutated residues with no coupling above 7 Å (*Schreiber and Fersht, 1995*) and that additivity is observed for distant mutations at rigid protein-protein interfaces (*Wells, 1990*). Our results agree with these findings since the residues in these YAP:TEAD pairs are distant from each other by more than 9 Å and TEAD is rigid and does not change conformation upon YAP binding.

We also measured $\Delta\Delta G_{int} < -0.5$ kcal/mol for one pair with residues belonging to each binding interface – hYAP Arg89:hTEAD4 Lys376 ($-0.62$ kcal/mol). This coupling energy, though moderate, is significant. Non-additivity in long-range interactions has also been observed in other systems (*Gianni et al., 2011*; *Jemth et al., 2014*; *LiCata and Ackers, 1995*; *Istomin et al., 2008*). This $\Delta\Delta G_{int}$ value is not linked to a specific protein preparation of hYAP Arg89 or hTEAD4 Lys376, as similar results were obtained with different batches of these proteins (data not shown). While they are located in two distant pockets at the YAP:TEAD interface, these two residues appear to act in a synergistic manner ($\Delta\Delta G_{int} < 0$) on the formation of the YAP:TEAD complex. There is no straightforward explanation for this result, but this pair is the only one that involves two charged residues (of the same charge) located at the two binding pockets. Since electrostatic interactions have a long-distance effect, an electrostatic component may explain the $\Delta\Delta G_{int}$ value measured between these

**Table 3.** Summary of the coupling energy ($\Delta\Delta G_{int}$) measured for each pair of residues. $\Delta\Delta G_{int}$ values are in kcal/mol and were calculated according to the description given in the Materials and methods section. Average and standard errors (SE) are shown. The gray cells indicate mutations of YAP and TEAD in the same binding pocket. n. m. (not measured) indicates that $\Delta\Delta G_{int}$ could not be determined experimentally because the $K_d$ values for the mutant YAP:mutant TEAD interaction in the corresponding double mutant cycles were above our assay limit (>200 µM).

| | | | hYAP[50-171] | | | | | | | | |
| --- | --- | --- | --- | --- | --- | --- | --- | --- | --- | --- | --- |
| | | | α-helix | | | Ω-loop | | | | | |
| | | | Leu65 | Leu68 | Phe69 | Met86 | Arg89 | Leu91 | Ser94 | Phe95 | Phe96 |
| hTEAD[217-434] | α-helix | Phe337 | −0.34 ±0.04 | −0.65 ±0.04 | −0.96 ±0.04 | 0.17 ±0.04 | −0.44 ±0.03 | n. m. | −0.09 ±0.05 | n. m. | 0.02 ±0.06 |
| | | Lys376 | −0.65 ±0.04 | −0.41 ±0.05 | −1.28 ±0.06 | 0.13 ±0.04 | −0.62 ±0.03 | −0.41 ±0.06 | −0.22 ±0.06 | −0.30 ±0.04 | −0.14 ±0.05 |
| | | Leu380 | −0.36 ±0.05 | −0.29 ±0.04 | −1.01 ±0.06 | 0.07 ±0.03 | −0.29 ±0.03 | −0.26 ±0.05 | −0.08 ±0.05 | −0.18 ±0.03 | −0.02 ±0.06 |
| | | Val389 | −0.52 ±0.06 | −0.29 ±0.05 | −1.11 ±0.05 | 0.16 ±0.03 | n. m. | n. m. | 0.03 ±0.05 | n. m. | −0.06 ±0.04 |
| | Ω-loop | Glu263 | −0.05 ±0.05 | −0.09 ±0.03 | −0.10 ±0.06 | 0.07 ±0.05 | −0.59 ±0.06 | −0.62 ±0.05 | −0.96 ±0.05 | −0.51 ±0.05 | −0.04 ±0.07 |
| | | Val265 | 0.26 ±0.03 | 0.15 ±0.03 | 0.13 ±0.05 | 0.09 ±0.03 | −0.58 ±0.02 | −0.28 ±0.06 | −0.06 ±0.06 | −0.61 ±0.04 | −0.22 ±0.06 |
| | | Asp272 | n. m. | −0.02 ±0.04 | n. m. | −1.85 ±0.04 | −3.49 ±0.05 | −2.74 ±0.06 | −1.65 ±0.05 | −2.50 ±0.07 | −1.67 ±0.06 |

amino acids. Using $\Delta\Delta G_{int}$ = −0.5 kcal/mol as the minimum value for significant energetic coupling between two residues, we calculated the $K_d$ ($K_d^{calc}$) of the mutant YAP:mutant TEAD interaction for the seven pairs for which no $\Delta\Delta G_{int}$ could be measured (n. m. in *Table 3*). For 3 of these pairs (hYAP Leu91:hTEAD4 Phe337, hYAP Phe95:hTEAD4 Phe337 and hYAP Leu65:hTEAD4 Asp272), we arrived at $K_d^{calc}$ <150 µM (*Supplementary file 2*), suggesting that if these residues had been coupled with $\Delta\Delta G_{int} \leq$ −0.5 kcal/mol, it would have been possible to measure the $K_d$ for the interaction between the two mutant proteins. For the remaining pairs – hYAP Phe69:hTEAD4 Asp272, hYAP Arg89:hTEAD4 Val389, hYAP Leu91:hTEAD4 Val389, and hYAP Phe95:hTEAD4 Val389 – it is not possible to draw a conclusion, since the $K_d^{calc}$ values are either just below or >200 µM, our upper limit for $K_d$ determination (*Supplementary file 2*).

We shall now look at the coupling energy between YAP and TEAD residues located in the same binding pocket. $\Delta\Delta G_{int}$ values larger than the experimental error limit were obtained with several pairs of residues. These values are all negative, indicating a favorable energetic coupling between these residues (*Table 3*). Within the α-helix, 5 $\Delta\Delta G_{int}$ values below −0.5 kcal/mol were measured for residues located in close proximity: hYAP Leu65:hTEAD4 Lys376 (−0.65 kcal/mol), hYAP Leu68: hTEAD4 Phe337 (−0.65 kcal/mol), hYAP Phe69:hTEAD4 Lys376 (−1.28 kcal/mol), Leu380 (−1.01 kcal/mol) and Val389 (−1.11 kcal/mol) (*Figure 3A*). These low to moderate $\Delta\Delta G_{int}$ values may reflect the van der Waals nature of the interactions between these residues, since such interactions usually give lower coupling energies (*Schreiber and Fersht, 1995*; *Goldman et al., 1997*). hYAP Phe69 is significantly coupled ($\Delta\Delta G_{int}$ < −0.95 kcal/mol) to the 4 TEAD residues studied (*Table 2*), highlighting once more the importance of this residue at the α-helix binding interface. Significant $\Delta\Delta G_{int}$ values were also measured between residues that are not directly in contact: hYAP Leu65:hTEAD4 Val389 (−0.52 kcal/mol) and hYAP Phe69:hTEAD4 Phe337 (−0.95 kcal/mol). Experimental bias (e.g. effect of the mutations on the unbound conformation of the proteins) cannot be ruled out, but, |$\Delta\Delta G_{int}$| values below 0.5 kcal/mol were measured in other pairs involving hYAP Leu65 or Phe69 and hTEAD4 Phe337 or Val389 (*Table 3*), suggesting the mutation of these residues does not systematically lead to high $\Delta\Delta G_{int}$ values. Therefore the residues of these two pairs are likely to be energetically coupled. $\Delta\Delta G_{int}$ is high (~1 kcal/mol) between the two hydrophobic residues hYAP Phe69 and hTEAD4 Phe337 that are not in direct contact. hYAP Phe69 and hTEAD4 Phe337, which are located

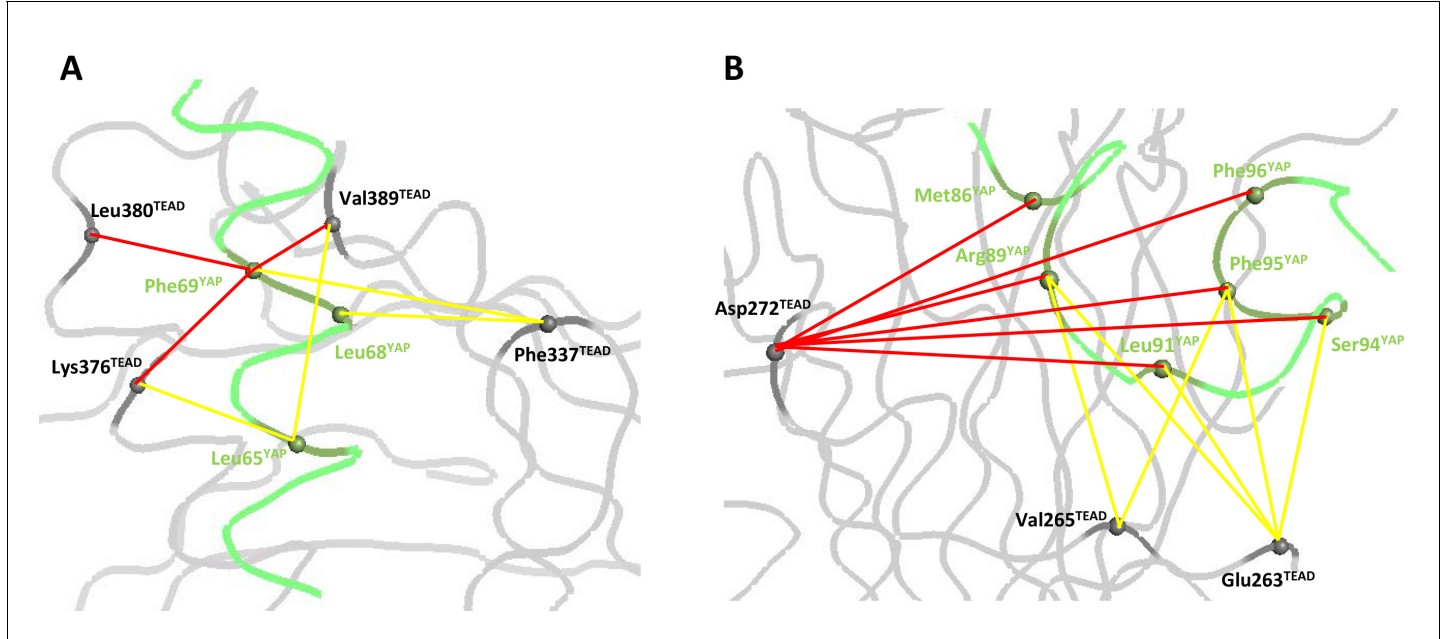

**Figure 3.** Coupling energies at the α-helix and Ω-loop binding pockets. (A) α-helix interface. (B) Ω-loop interface. The gray and green ribbons represent the main polypeptide chain of TEAD and YAP, respectively. The $C_\alpha$ of the mutated amino acids are represented by spheres. The coupling energies ($\Delta\Delta G_{int}$) between the different residues are symbolized by lines. Yellow lines: $-1$ kcal/mol $< \Delta\Delta G_{int} < -0.5$ kcal/mol. Red lines: $\Delta\Delta G_{int} < -1$ kcal/mol.

at each side of the bound α-helix (*Figure 4—figure supplement 1*), may act synergistically in holding/orienting the hYAP α-helix in its correct bound orientation.

In the Ω-loop interface, 12 $\Delta\Delta G_{int}$ values below $-0.5$ kcal/mol were measured (*Table 3*, *Figure 3B*). Significant coupling ($-0.96$ kcal/mol) was observed between hYAP Ser94 and hTEAD4 Glu263, two residues within hydrogen bond distance (*Figure 1D*). hYAP Arg89, Leu91, and Phe95 are also coupled to hTEAD4 Glu263, albeit to a lesser extent. Since these 3 YAP residues are not in contact with hTEAD4 Glu263, coupling might be indirect and mediated via hYAP Ser94. Secondary coupling via a third side chain was previously described in other systems (*Schreiber and Fersht, 1995*). Moderate $\Delta\Delta G_{int}$ values were also obtained between hTEAD4 Val265 and hYAP Arg89 ($-0.58$ kcal/mol) and Phe95 ($-0.61$ kcal/mol) residues that are also not in direct contact. The largest coupling energy was observed between hYAP Arg89 and hTEAD4 Asp272, $\Delta\Delta G_{int} = -3.49$ kcal/mol (*Table 3*). The double mutant cycle shows that, once hYAP Arg89 is mutated, the hTEAD4 Asp272Ala mutation has little effect on binding ($\Delta\Delta G_4 = 0.04$ kcal/mol, *Figure 4A*). This is in agreement with the structural data showing that hYAP Asp272 interacts only with hYAP Arg89 (*Figure 1C*). The larger effect on binding of the hYAP Arg89Ala mutation against an hTEAD4 Asp272Ala background ($\Delta\Delta G_2 = 0.85$ kcal/mol, *Figure 4A*) can be explained by the loss of both additional intermolecular interactions with TEAD (*Figure 1C*) and intramolecular stabilizing interactions in the bound YAP (*Hau et al., 2013*). hTEAD4 Asp272 is also strongly coupled ($\Delta\Delta G_{int} < -1.6$ kcal/mol, *Table 3*) to each of the hYAP Ω-loop residues studied. This effect is limited to these amino acids, since no significant coupling energy was measured between hTEAD4 Asp272 and hYAP Leu68, which is located in the α-helix region ($\Delta\Delta G_{int} = -0.02$ kcal/mol, *Table 3*). These hydrophobic/polar Ω-loop residues are therefore indirectly coupled to hTEAD4 Asp272 via hYAP Arg89, because they make no direct contact with this amino acid. The hYAP Arg89:hTEAD4 Asp272 interaction is thus crucial for the binding of the Ω-loop, since it allows several residues to act in a synergistic manner on the formation of the YAP:TEAD complex. Several proline residues – hYAP Pro85, Pro92, Pro98, and Pro99 – are also present in the Ω-loop, where they play a role in complex formation (*Hau et al., 2013*; *Zhang et al., 2014*). However, these residues are difficult to study with the

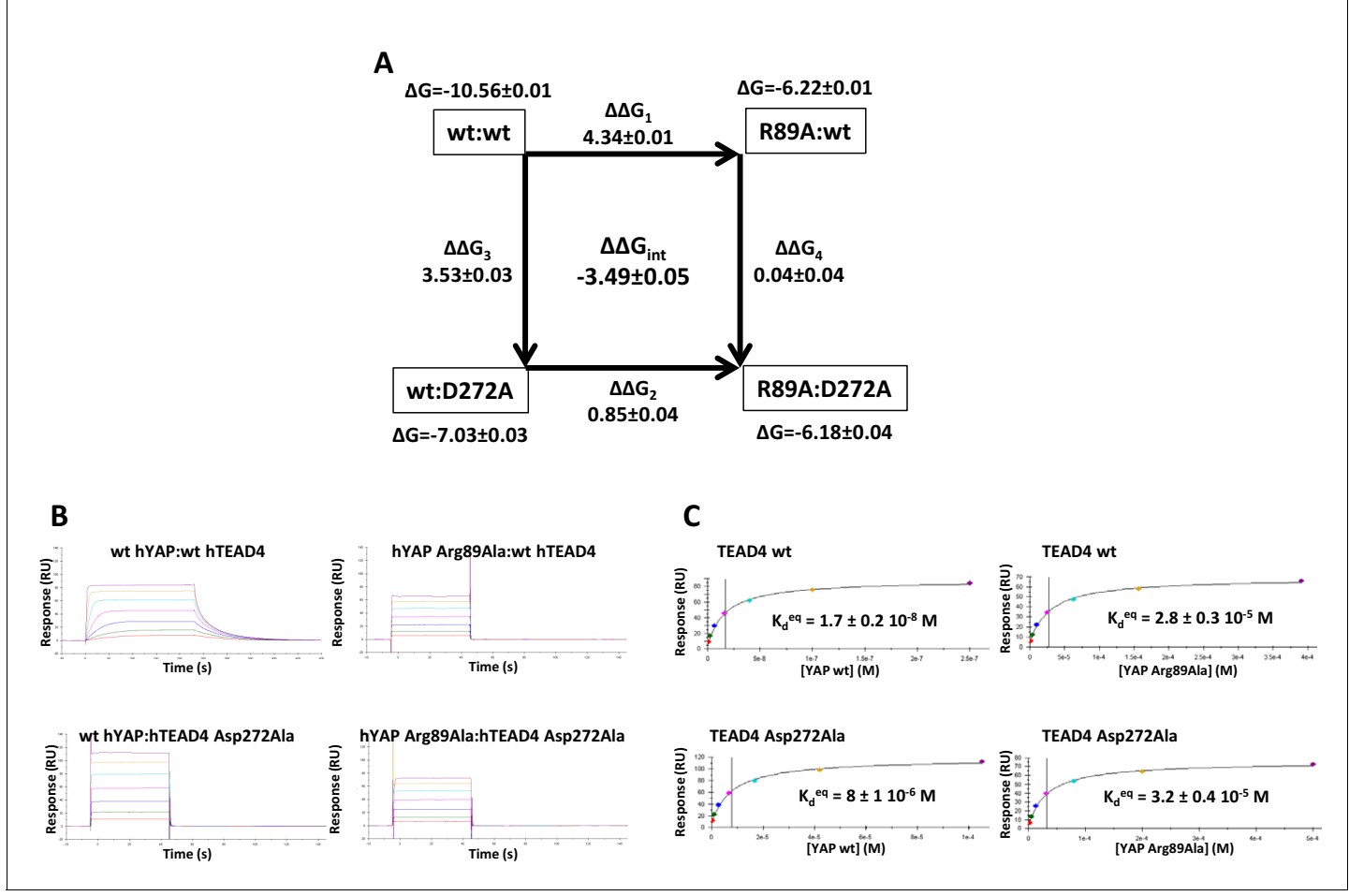

**Figure 4.** Double mutant cycle for the hYAP Arg89:hTEAD4 Asp272 interaction. (**A**) The binding energies (kcal/mol) measured for the different pairwise interactions are indicated in boxes, and the differences between these binding energies ($\Delta\Delta G_{1-4}$) are shown. The coupling energy ($\Delta\Delta G_{int} = \Delta G_{wt:wt} + \Delta G_{R89A:D272A} - \Delta G_{R89A:wt} - \Delta G_{wt:D272A}$) is indicated (kcal/mol). Standard errors (SE) are given. (**B**) Representative sensorgrams of the Surface Plasmon Resonance experiments carried out to establish the double mutant cycle presented on **A**. wt or Asp272Ala hTEAD4 were immobilized on sensor chips and the binding of different concentrations of analyte (hYAP wt or Arg89Ala) was measured. $K_d$ values were measured at equilibrium ($K_d^{eq}$). The contact time in the experiments was varied according to the affinity of each of the YAP proteins. (**C**) Binding isotherms obtained from the sensorgrams presented on **B**. The experiments were fitted with the Biacore T200 evaluation software using a one site binding with background model. The $K_d^{eq}$ values and the standard error (SE) from the fit are indicated.

The following source data and figure supplement are available for figure 4:

**Source data 1.** The double mutant cycles established in this study.
**Figure supplement 1.** Position of hYAP Phe69 and hTEAD4 Phe337 at the α-helix binding pocket.

techniques used in this report, because their mutation may affect the local structure of YAP in addition to disrupting interactions with TEAD.

In conclusion, we have studied the interaction between YAP and TEAD via single site-directed mutagenesis and multiple DMCAs. Our analysis, which was focused on a subset of key residues located at the YAP:TEAD interface, provides the first quantitative mapping on the individual contribution of these amino acids to the formation of the YAP:TEAD complex. We show that hYAP Phe69 is a 'hot-spot' residue at the α-helix interface. We find that hYAP Arg89 and hTEAD4 Lys376, which are not located at the same binding interface, act in a synergistic manner in the formation of the YAP:TEAD complex. We reveal that the hYAP Arg89:hTEAD4 Asp272 interaction is essential for the interaction between the Ω-loop region of YAP and TEAD. We provide evidence for the key role of

hTEAD4 Asp272 in the interaction between YAP and TEAD both *in vitro* and in a cellular context. Overall, our study provides a set of new information on the YAP:TEAD interaction, which is a key effector node in the Hippo pathway.

## Materials and methods

### Cloning, expression, purification and analysis

The MCAT_Luc reporter construct in pLENTI6TR backbone has been described (*Michaloglou et al., 2013*). The derivation of the stable HEK293T::MCAT_Luc model by lentiviral transduction was conducted as previously described for other cell lines (*Michaloglou et al., 2013*); a representative clone (#11) was used in this study. The wt hYAP construct (*Michaloglou et al., 2013*) was cloned into Gateway-compatibilized pcDNA3.1_hygro (Invitrogen, Carlsbad, CA) according to the manufacturer's protocol. The pCMVSport6::TEAD4 full-length cDNA clone was obtained from imaGenes (now Source BioScience, United Kingdom) and transferred by standard PCR and consecutive Gateway reactions into pcDNA3.1/nV5-DEST expression vector (Invitrogen, Carlsbad, CA). The Asp272Ala mutation in hTEAD4 was introduced by use of the QuikChange Lightning Site-Directed Mutagenesis kit (Agilent Technologies, Germany) and sequence-verified.

Human N-Avitagged hTEAD4$^{217-434}$ protein was cloned, expressed and purified as previously described (*Hau et al., 2013*). The purified hTEAD4 protein is 100% acylated at hTEAD4 Cys367. YAP$^{50-171}$ (NM_001130145.2) with an N-terminal hexa-histidine-tag followed by a HRV 3C protease cleavage site was expressed from a pACYCDuet-1 vector. Mutations in TEAD4 and YAP proteins were introduced with the QuikChange II Lightning site-directed mutagenesis kit (Agilent Technologies, Germany) according to the manufacturer's instructions and confirmed by Sanger sequencing.

For expression of the YAP proteins, a pre-culture of LB medium containing 34 μg/ml chloramphenicol was inoculated with *Escherichia coli* NiCo21 (DE3) cells (New England Biolabs, Ipswich, MA) transformed with the expression plasmid and grown overnight at 37°C. A 1:1 mixture of LB and TB medium supplemented with 50 mM MOPS and chloramphenicol was inoculated with the pre-culture. At $OD_{600}$ = 0.8 the culture was chilled to 18°C, and the protein expression was induced by addition of 0.2 mM IPTG and run overnight. Bacterial cells were harvested by centrifugation at 6000 g for 20 min and frozen on dry ice. Cell pellets were thawed and suspended in 50 mM TRIS.HCl, 300 mM NaCl, 30 mM imidazole, pH 7.8 supplemented with Complete Protease Inhibitor (Roche, Switzerland) and Benzonase (Merck, Germany). The cells were then mechanically lysed by an EmulsiFlex C3 homogenizer (Avestin, Canada). Insoluble cell debris was removed by centrifugation for 50 min at 48000 g. The clarified cell lysate was loaded onto a 5 ml HisTrap HP column (GE Healthcare, United Kingdom) mounted on an ÄKTA Pure system (GE Healthcare, United Kingdom) and the column washed with 10 column volumes of 50 mM TRIS.HCl, 300 mM NaCl, 30 mM imidazole, pH 7.8. The YAP protein was proteolytically cleaved from the bound affinity tag by GST-tagged HRV 3C protease overnight at 4°C. YAP was eluted with wash buffer and dialyzed overnight at 5°C against an excess of 20 mM PIPES, 20 mM NaCl, 0.1 mM TCEP, pH 6.1 (Buffer A). The dialyzed protein was then loaded onto a 1 ml Resource S column (GE Healthcare, United Kingdom) and eluted with a linear gradient of Buffer A with 300 mM NaCl. The protein was pooled and concentrated with Amicon Ultra 4 Ultracell 3K columns (Millipore, Billerica, MA) and loaded onto a Superdex 75 10/300 GL size exclusion column (GE Healthcare, United Kingdom) equilibrated with 50 mM HEPES.NaOH, 100 mM KCl, 0.25 mM TCEP, 1 mM EDTA, 0.05% (v/v) Tween 20. Pure protein was finally concentrated to about 10 mg/ml in an Amicon concentrator. The final yield of pure protein was between 3 and 5 mg per liter expression culture.

The purity and the molecular weight of all the purified proteins were assessed by LC-MS. The concentration of the different protein preparations was determined by reverse phase (RP) HPLC measuring the absorbance at 210 nm and using calibration curves made with BSA.

### Cell culture and transfections

HEK293T (RRID:CVCL_0063) and HEK293FT (RRID:CVCL_6911) cell lines were obtained from Sigma-Aldrich (Saint Louis, MO) and Invitrogen (Carlsbad, CA)/ThermoFisher Scientific (Waltham, MA), respectively. The identity of cell lines was authenticated by internal SNP genotype profiling. The absence of mycoplasma contamination was regularly verified (Venor GeM Mycoplasma PCR

Detection kit, Minerva Biolabs, Germany). Both cell lines were maintained in DMEM supplemented with 10% (v/v) fetal calf serum (AMIMED, United Kingdom), 2 mM L-glutamine, 1 mM sodium pyruvate and 0.1 mM MEM non-essential amino acids. Transient transfections were performed with a DNA mix containing the plasmids of interest using Lipofectamine 2000 (Invitrogen, Carlsbad, CA), according to the manufacturer's protocol.

## YAP/TEAD Immunoprecipitation (IP)

For IP experiments, HEK293FT cells were transfected with 500 ng each of YAP and nV5-TEAD4 cDNA constructs, and lysed in RIPA buffer (derived from 10x stock (Millipore, Billerica, MA); final concentration of components: 50 mM TRIS.HCl pH 7.2, 120 mM NaCl, 1% Nonidet P40 (v/v), 1 mM EDTA and 0.1% (v/v) SDS; supplemented with 2 mM sodium orthovanadate, 6 mg/ml sodium pyrophosphate and PhosSTOP and Protease Inhibitor Cocktail (both from Roche, Switzerland)) 48 hr afterwards. Lysates (250 µg) were then incubated with YAP1 antibody overnight under rotation at 4°C, followed by incubation with Dynabeads Protein G (Invitrogen, Carlsbad, CA) for 2 hr under rotation at 4°C. Immunoprecipitates were washed three times with RIPA buffer lacking SDS, eluted with Laemmli Sample Buffer (BioRad, Hercules, CA) by incubation at 95°C for 5 min and resolved by standard SDS-PAGE gel electrophoresis and Western Blotting. The following antibodies were used. For IP: YAP1 (EP1674Y; Abcam (United Kingdom),ab52771). For Western Blot analysis: YAP1 (D8H1X XP; Cell Signaling Technology (Danvers, MA), #14074) and V5 (Invitrogen (Carlsbad, CA), R96025) as primary antibodies; HRP-anti-rabbit (Cell Signaling Technology (Danvers, MA), #7074) and HRP anti-mouse ( GE Healthcare (United Kingdom), NA931V) as secondary antibodies.

## Luciferase assay

HEK293T cells stably expressing the MCAT_Luc reporter (clone #11) were transfected with 100 ng each of YAP and nV5-TEAD4 cDNA constructs, and processed for lysate derivation and luciferase assay as described in *Michaloglou et al. (2013)*. All luciferase readings were normalized to resazurin and are depicted as the average of 3 independent experiments ± STDEV.

## Fluorescence-based thermal denaturation assay

The N-Avitagged hTEAD4$^{217-434}$ proteins were diluted at 2 µM in assay buffer (50 mM HEPES, 100 mM KCl, 0.25 mM TCEP, 1 mM EDTA, 2% (v/v) DMSO, pH 7.4) in the presence of 2x SYPRO Orange dye (ThermoFisher Scientific, Waltham, MA). This mix was added to 384-well, thin-walled Hard-Shell PCR microplates (BioRad, Hercules, CA). Before reading, the plates were covered by optically clear adhesive seals. Measurements were carried out with a CFX384 Real-Time PCR Detection System (BioRad, Hercules, CA). The temperature was increased from 25°C to 85°C at 0.5°C / 30 s and the fluorescence intensity was measured with the excitation and emission filters set to 465 and 590 nm, respectively. The melting temperatures ($T_m$) were determined by analyzing the thermal denaturation curves with CFX Manager (BioRad, Hercules, CA).

## Surface Plasmon Resonance (SPR)

All the experiments were carried out with a Biacore T200 optical biosensor and Series S sensor Chip SA (Biacore AB, GE Healthcare, United Kingdom). The chips were washed three times with 1 M NaCl/50 mM NaOH, and the N-Avitagged hTEAD4$^{217-434}$ proteins were injected at a flow rate of 5 µl/min in SPR immobilization buffer (50 mM HEPES, 100 mM KCl, 0.25 mM TCEP, 1 mM EDTA, 0.05% (v/v) Tween 20, 0.05% (w/v) BSA, pH 7.4). Kinetic experiments were performed at 25°C with a flow rate of 50 µl/min in SPR running buffer (SPR immobilization buffer containing 2% (v/v) DMSO). The YAP proteins were diluted in SPR running buffer. After baseline equilibration with a series of buffer blanks, a DMSO correction series was performed from 1% to 3%. Each cycle consisted of an injection phase of YAP$^{50-171}$ (25 to 230 s) and a dissociation phase (75 to 450 s). All data were referenced for a blank streptavidin reference surface and blank injections of running buffer to minimize the influence of baseline drift upon binding.

A 6-step workflow was applied to obtain the dissociation constants measured at equilibrium ($K_d^{eq}$). Steps 1 to 3 ensure that only high-quality sensorgrams are selected for $K_d^{eq}$ determination. Steps 4 to 6 ensure that $K_d^{eq}$ is measured in experimental conditions at or near to TEAD saturation.

Step 1. The sensorgrams were globally fitted with a 1:1 interaction model using the Biacore T200 evaluation software.

Step 2. Each sensorgram was visually inspected and low-quality experiments (e.g. large bulk effects, unstable signal at equilibrium, inappropriate YAP concentration range, etc.) were not further analyzed.

Step 3. Experiments with a standard error on fitted $K_d^{eq}$ higher than 20% $K_d^{eq}$ or a fitted maximum binding capacity ($R_{max}^{fitted}$) <70% theoretical $R_{max}$ ($R_{max}^{theo}$) ($R_{max}^{theo} = (MW^{YAP}/ MW^{TEAD})$ x $R_{TEAD}$ x n with $MW^{YAP}$ and $MW^{TEAD}$ molecular weight of YAP and TEAD4, respectively, $R_{TEAD}$ level of immobilization of TEAD4 in response unit (RU) and in stoichiometry here n = 1) were discarded.

Step 4. $K_d^{eq}$ values from experiments meeting criteria of steps 1–3 for which at least the highest YAP concentration used in the dose response was >10 $K_d^{eq}$ (saturation or near saturation conditions) are reported in the text.

Step 5. For the experiments that did not meet step four criteria, $R_{max}$ at saturation ($R_{max}^{sat}$) were calculated. $R_{max}^{sat}$ values correspond to the signal that should be measured for a YAP:TEAD pair when all the reactive immobilized TEAD molecules are bound to YAP (saturation conditions). $R_{max}^{sat}$ = $R_{max}^{theo}$ x ($R_{max}^{meas*}$ / $R_{max}^{theo*}$). $R_{max}^{theo}$ values were calculated for each YAP:TEAD pair as defined above. $R_{max}^{theo*}$ and $R_{max}^{meas*}$ values were obtained from measurements done with exactly the same TEAD protein preparation but using wt $hYAP^{50-171}$ as analyte. The ratio $R_{max}^{meas*}$ / $R_{max}^{theo*}$ is an experimental estimate of the fraction of TEAD available for binding in conditions where saturation is achieved, since it was always obtained with wt $hYAP^{50-171}$ used as analyte. Experiments for which the signal measured at equilibrium ($R_{eq}$) at the highest YAP concentration (500 µM) did not exceed 75% of $R_{max}^{sat}$ were considered not suitable for accurate $K_d^{eq}$ determination because they were too far from saturation conditions. These experiments are not reported.

Step 6. For the experiments meeting step five criteria, the $R_{eq}$ determined at different YAP concentrations with the Biacore T200 evaluation software were fitted using a one site binding with background equation with Prism v7 (GraphPad Software) including in the analysis the additional point: [YAP] = 50 $K_d^{eq}$ (estimated from initial global fits of the sensorgram); $R_{eq}$ = $R_{max}^{sat}$. Simulations made on several experimental data sets show that the inclusion of this point significantly improves the accuracy of the fit, and the standard error of the fitted $K_d^{eq}$ was always lower than 10% $K_d^{eq}$. The dissociation constants determined under such conditions are reported in the text.

Within our experimental conditions, the limit for accurate $K_d^{eq}$ determination is 200 µM. All the pairs with a lower affinity are reported in the text as $K_d^{eq}$ > 200 µM.

## Double mutant cycle analysis

The double mutant cycles were constructed measuring $K_d^{eq}$ for four different pairwise interactions: wtYAP:wtTEAD (wY:wT), mutantYAP:wtTEAD (mY:wT), wtYAP:mutantTEAD (wY:mT) and mutantYAP:mutantTEAD (mY:mT). The binding-free energies were calculated from $K_d^{eq}$ with the formula $\Delta G = RTLn\, K_d^{eq}$ (R = 1.986 cal.mol$^{-1}$K$^{-1}$, T = 298°K). The interaction energy between two residues, $\Delta\Delta G_{int}$, were calculated according to the formula $\Delta\Delta G_{int} = \Delta G_{mY:mT} + \Delta G_{wY:wT} - \Delta G_{mY:wT} - \Delta G_{wY:mT}$. Standard errors (SE) on $\Delta\Delta G_{int}$ were derived from $SE_{\Delta\Delta Gint} = (SE_{\Delta GmY:mT}^2 + SE_{\Delta GwY:wT}^2 + SE_{\Delta GmY:wT}^2 + SE_{\Delta GwY:mT}^2)^{1/2}$, where SE is the standard error. Large experimental errors in $K_d^{eq}$ determination are a source of considerable uncertainty on $\Delta\Delta G_{int}$ because of error propagation in the calculations. As a consequence, only high $\Delta\Delta G_{int}$ values may become significant (*Pons et al., 1999*). Within our experimental conditions, the SE for $K_d^{eq}$ did not exceed 15% $K_d^{eq}$ (see *Tables 1* and *2* and *Supplementary file 1*). Since $SE_{\Delta G}$ = RT ($SE_{Kdeq}$ / $K_d^{eq}$), the maximum $SE_{\Delta G}$ = 0.09 kcal/mol and as a consequence the maximum $SE_{\Delta\Delta Gint}$ is 0.18 kcal/mol. Using a 2SE threshold (95% confidence interval), $|\Delta\Delta G_{int}|$ > 0.36 kcal/mol should be significant. However, we shall only consider $|\Delta\Delta G_{int}|$ > 0.5 kcal/mol to be high enough to reflect an energy coupling between two residues.

## Acknowledgements

We thank E Billy, H Niu and ME Digan (Novartis) for generating the HEK293T::MCAT_Luc reporter model and A Leu (Novartis) for derivation of the nV5-TEAD4 construct. We thank J Groarke (Novartis) for generating the pACYCDuet-1 vector.

## Additional information

### Competing interests

FB, MM, PF, CZ, TM, CD, DE, TS, PC: The research was funded by Novartis, Inc., where all authors were employees at the time the study was conducted. The authors declare no other competing financial interests. The other author declares that no competing interests exist.

### Funding

| Funder | Author |
| --- | --- |
| Novartis | Yannick Mesrouze |
| | Fedir Bokhovchuk |
| | Marco Meyerhofer |
| | Patrizia Fontana |
| | Catherine Zimmermann |
| | Typhaine Martin |
| | Clara Delaunay |
| | Dirk Erdmann |
| | Tobias Schmelzle |
| | Patrick Chène |

The funders had no role in study design, data collection and interpretation, or the decision to submit the work for publication.

### Author contributions

YM, MM, CZ, TM, CD, Data curation, Formal analysis, Investigation, Methodology; FB, PF, Investigation, Methodology; DE, TS, Conceptualization, Data curation, Formal analysis, Supervision, Investigation, Methodology; PC, Conceptualization, Data curation, Formal analysis, Supervision, Validation, Investigation, Visualization, Methodology, Writing—original draft, Writing—review and editing

### Author ORCIDs

Patrick Chène, http://orcid.org/0000-0002-6010-9169

## Additional files

### Supplementary files

• Supplementary file 1. Dissociation constants for the different YAP:TEAD complexes. The N-Avi-tagged $hTEAD4^{217-434}$ proteins were immobilized on sensor chips and their affinity for the $hYAP^{50-171}$ proteins was measured at 298°K by Surface Plasmon Resonance in $n \geq 3$ independent experiments. $K_d$ values (in nM) were obtained from equilibrium data ($K_d^{eq}$). Averages and standard errors (SE) are given.

• Supplementary file 2. Calculated $K_d$ for pairs of hYAP and hTEAD4 mutants. $K_d$ values were calculated ($K_d^{calc}$) assuming $\Delta\Delta G_{int} = -500$ cal/mol between the YAP:TEAD pairs. A calculation example for the hYAP Arg89:hTEAD4 Val389 pair is given in the following. The difference in binding energy between the two interactions wt hYAP:wt hTEAD4 and hYAP Arg89Ala:wt hTEAD4 is $\Delta\Delta G = 4338$ cal/mol (*Figure 4—source data 1*, $\Delta\Delta G_1$ double mutant cycle for hYAP Arg89:hTEAD4 Val389). With $\Delta\Delta G_{int} = -500$ cal/mol, the difference in binding energy between the interactions wt hYAP:hTEAD4 Val389Ala and hYAP Arg89Ala:wt hTEAD4 Val389Ala should be $\Delta\Delta G = 3838$ cal/mol. As $\Delta G = -8881$ cal/mol for wt hYAP:hTEAD4 Val389Ala, the binding energy of the interaction between hYAP Arg89Ala and wt hTEAD4 Val389Ala should be $\Delta G = -5043$ cal/mol. Using $K_d^{calc} = e^{\Delta G/RT}$ the dissociation constant for the interaction hYAP Arg89Ala:wt hTEAD4 Val389Ala should be $K_d^{calc} = 199$ μM. The last column gives the calculated $K_d$ values for the interaction between the two mutant proteins ($K_d^{add}$) if no coupling exists between the two residues ($\Delta\Delta G_{int} = 0$ cal/mol in the above calculation). All $K_d^{add}$ values are above 200 μM in agreement with our experimental data.

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
