## [Decision Letter]

Thank you for submitting your article "Dissection of the interaction between the intrinsically disordered YAP protein and the transcription factor TEAD" for consideration by *eLife*. Your article has been favorably evaluated by Richard Aldrich (Senior Editor) and three reviewers, one of whom, Jane Clarke (Reviewer #1), is a member of our Board of Reviewing Editors. The following individual involved in review of your submission has agreed to reveal their identity: Tim Sharpe (Reviewer #2).

The reviewers have discussed the reviews with one another and the Reviewing Editor has drafted this decision to help you prepare a revised submission.

The authors present a meticulous biophysical dissection of the interaction between human YAP and TEAD4, predominantly using protein engineering and surface plasmon resonance. The study identifies residues from both proteins that are energetically important for binding at the interface of hTEAD with both the α-helical and Ω-loop regions of hYAP.

Summary:

The reviewers all thought the study was well executed and of general interest. They were enthusiastic and recommended acceptance. The points below are revisions you should make to clarify the manuscript, make it easier to read, and easier for the more general, less "biophysical" reader to appreciate.

Essential revisions:

1) You might like to consider the way things are presented. The manuscript is dense and the results are hard to follow: Use of the "Results and Discussion" format is not ideal here. The details of what interactions being probed can then be left to the detailed discussion of the results and the general picture of the mode of binding can be discussed later.

2) It would be nice to have a structural figure showing some of the hTEAD residues that have been mutated in their structural context. And a figure showing the whole bound YAP peptide, so that the whole conformation can better be visualised without needing to refer to the structure. It might be useful to make a figure in which the ddGint data in Table 3 are presented in the context of the structure, perhaps with some kind of color code to visualize the coupling energies.

3) Some discussion comparing the results of such a double mutant cycle analysis of IDP:protein with that of two folded proteins would be of interest. (and make the paper more generally interesting to the audience. Perhaps consider issues of the size of interfaces, the relative burial of the key residues, the cooperativity detected…

4) In this context one reviewer comments "Subsection “Double mutant cycle analyses”, third paragraph: perhaps the context is different here, since YAP is disordered before binding. Is it possible that either the α-helical region or Ω-loop region could completely dissociate from the interface with sufficient destabilisation and become disordered, while the other site remained bound? Would this have different entropic consequences (e.g. end effects) than one would observe for two spatially-separated contact points between folded proteins? This is only a philosophical point, since it appears that there's no significant coupling between the sites apart from electrostatics. A study of the unbound and bound-state conformational dynamics of selected YAP mutants by NMR might be an interesting follow-up (definitely not required for this paper though)."

5) In the Discussion you say: "On the basis of this finding, a binding mechanism can be proposed for the Ω-loop." And then go on to propose such a mechanism. It may be tempting to do this, but as I know you are aware it is not possible to propose a mechanism without kinetics. This should be removed.

---

## [Author Response]

Essential revisions:

1) You might like to consider the way things are presented. The manuscript is dense and the results are hard to follow: Use of the "Results and Discussion" format is not ideal here. The details of what interactions being probed can then be left to the detailed discussion of the results and the general picture of the mode of binding can be discussed later.

I have tried to change the manuscript to create a Results and a Discussion section but I failed to draft a version that I found suitable. I apologize for this. Our study is a structure-function analysis and it is difficult to uncouple the results from their immediate analysis/interpretation. When I try to write the manuscript as suggested in the review “The details of what interactions being probed can then be left to the detailed discussion of the results” and afterward “the general picture of the mode of binding”, I repeat in the Discussion section several observations/comments already made in the Results section. So, the article is becoming repetitive and it does not gain in clarity.

2) It would be nice to have a structural figure showing some of the hTEAD residues that have been mutated in their structural context. And a figure showing the whole bound YAP peptide, so that the whole conformation can better be visualised without needing to refer to the structure. It might be useful to make a figure in which the ddGint data in Table 3 are presented in the context of the structure, perhaps with some kind of color code to visualize the coupling energies.

Two new figures have been inserted in the text. Figure 1 represents the TEAD residues mutated in the α-helix pocket and for which a K_d_ has been measured. Figure 1 represents the TEAD residues mutated in the Ω-loop pocket and for which a K_d_ has been measured. Since the α-helix and the Ω-loop interfaces are distant from each other, it is not possible to make a single figure showing the interactions in both pockets at the same time. The text and the legend of Figure 1 have been updated.

We also introduced a new figure, Figure 3. This figure represents the coupling energies between residues at the two binding interfaces. This is a good suggestion from the reviewers because it gives a summary of our data in a visual manner easier to interpret than the table in the text. This figure may help the reader in getting a general picture of the findings presented in the manuscript (see point 1 above).

3) Some discussion comparing the results of such a double mutant cycle analysis of IDP:protein with that of two folded proteins would be of interest. (and make the paper more generally interesting to the audience. Perhaps consider issues of the size of interfaces, the relative burial of the key residues, the cooperativity detected.

This topic is interesting and could the subject of a future review article. However, we do not think that sufficient double mutant cycle analyses on IDP:protein interactions are available today in the literature to make such comparison and to lead to convincing conclusions. For example, most of the ΔGint measured by Jemth et al. (J. Biol. Chem. 289 5528, 2014) for the interaction between NCBD (molten globule) and ACTR (IDP) are positive while they are negative for YAP:TEAD. This already shows differences amongst IDPs suggesting that comparisons between different systems might be challenging.

4) In this context one reviewer comments "Subsection “Double mutant cycle analyses”, third paragraph: perhaps the context is different here, since YAP is disordered before binding. Is it possible that either the α-helical region or Ω-loop region could completely dissociate from the interface with sufficient destabilisation and become disordered, while the other site remained bound? Would this have different entropic consequences (e.g. end effects) than one would observe for two spatially-separated contact points between folded proteins? This is only a philosophical point, since it appears that there's no significant coupling between the sites apart from electrostatics. A study of the unbound and bound-state conformational dynamics of selected YAP mutants by NMR might be an interesting follow-up (definitely not required for this paper though)."

We do not have experimental data to answer this question, but, some of our previous results ([Hau et al., 2013] in the manuscript) show that peptides mimicking the isolated α-helix and the Ω-loop of YAP have an affinity >150 and ~70 μM, respectively. This suggests that if the α-helix or the Ω-loop is mutated such that it can completely dissociate from the interface, the remaining site will have alone a very weak affinity for TEAD and most probably it will dissociate also from the interface. We therefore speculate that if a mutation prevents one interface to be formed the complex will not be created or will be highly unstable.

5) In the Discussion you say: "On the basis of this finding, a binding mechanism can be proposed for the Ω-loop." And then go on to propose such a mechanism. It may be tempting to do this, but as I know you are aware it is not possible to propose a mechanism without kinetics. This should be removed.

Yes we agree: binding mechanisms cannot be established without the support of kinetic data. We even wrote this in the submitted manuscript: “Binding kinetic experiments may help to determine experimentally whether the interaction between YAP and TEAD follows such a mechanism or not.” We are surprised that the reviewers did not take into consideration this warning sentence when making their comment. The misunderstanding may come from our phrasing. Instead of “On the basis of this finding, a binding mechanism can be proposed for the Ω-loop.” we should have written: “On the basis of this finding, one could hypothesize a binding mechanism for the Ω-loop.” We think that scientific articles should (and many do) contain hypotheses/models that trigger future activities to prove or disprove them. In our case the mapping of the transition state could be the way to check the proposed mechanism. Of course, the technical feasibility of such experiments remains to be demonstrated since many of our mutant proteins have high K_d_. This is why we wrote in our warning sentence: “Binding kinetic experiments *may* help …” We feel that this paragraph gives perspectives beyond the current work and makes the link between the current thermodynamic data and future (if technically feasible) kinetic measurements.

Nevertheless we – sadly – followed the recommendation of the reviewers and removed this paragraph from the manuscript. The word “mechanistic” was removed from the Abstract. The sentence “Finally, our data provide an insight into the interactions taking place at the Ω-loop binding pocket, suggesting a binding mechanism for this secondary structure element.” was also removed from the conclusion.